# Study on the Effect of Non-Symmetrical Current Distribution Controlled by Capacitor Placement in Radio-Frequency Coils for 7T MRI

**DOI:** 10.3390/bios12100867

**Published:** 2022-10-12

**Authors:** Daniel Hernandez, Taewoo Nam, Yonghwa Jeong, Donghyuk Kim, Kyoung-Nam Kim

**Affiliations:** 1Neuroscience Research Institute, Gachon University, Incheon 21988, Korea; 2Department of Health Sciences and Technology, GAIHST, Gachon University, Incheon 21999, Korea; 3Department of Biomedical Engineering, Gachon University, Seongnam 13120, Korea

**Keywords:** current distribution, electromagnetic (EM), radiofrequency (RF), magnetic resonance imaging (MRI)

## Abstract

In this paper, we present a study on the effects of varying the position of a single tuning capacitor in a circular loop coil as a mechanism to control and produce non-symmetric current distribution, such that could be used for magnetic resonance imaging (MRI) operating at ultra-high frequency (UHF). This study aims to demonstrate that the position of the tuning capacitor of a circular loop could improve the coupling between adjacent coils, used to optimize transmission field uniformity or intensity, improve signal-to-noise ratio (SNR) or specific absorption rate (SAR). A typical loop coil used in MRI consists of symmetrically distributed capacitors along the coil; this design is able to produce uniform current distributions inside the coil. However, in UHF conditions, the magnetic flux density (|B_1_^+^|) field produced by this setup may exhibit field distortion, requiring a method of controlling the field distribution and improving the field intensity of the circular loop coil. The control mechanism investigated in this study is based on the position of the tuning capacitor in the circular coil, the capacitor position was varied from 15° to 345°, in steps of 15°. We performed electromagnetic (EM) simulations, fabricated the coils, and performed MRI experiments at 7T, with each of the coils with capacitor position from 15° to 345° to determine the effects on field intensity, coupling between adjacent coils, SAR, and applications for field uniformity optimization. For the case of free space, a coil with capacitor position at 15° showed higher field intensity compared to the reference coil; while an improved decoupling was achieved when a coil had the capacitor placed at 180° and the other coil at 90°; in a similar matter, we discuss the results for SAR, field uniformity and an application with an array coil for the spinal cord.

## 1. Introduction

The use of ultra-high-field magnetic resonance imaging (MRI) is desirable because of its capability of producing images with a higher signal-to-noise ratio (SNR) [1,2]. In principle, this is possible because more spins align in the presence of a stronger magnetic field, producing a stronger signal. When the main magnetic field strength is increased, the frequency at which the spins resonate also increases, as expressed by the Larmor frequency equation [3]. Therefore, the frequencies of operation of the transmission and reception of radiofrequency (RF) devices are also increased. However, increasing the frequency of operation signifies that the wavelength is reduced, which, in combination with the frequency dependency of the electrical properties of materials [4,5,6], reduces the effectiveness of the RF transmission (|B_1_^+^|) field inside the imaging target [7,8,9,10,11]. This results in field non-uniformity and the production of local hot spots of absorbed energy, as measured by the specific absorption rate (SAR) [12,13,14].

Creating an RF coil with uniform and low SAR is a challenge that has constantly been a topic of research. Several RF transmission and reception coils [15,16,17] have been proposed to address the occurrence of field non-uniformity. The use of high frequencies allows the application of antennas such as microstrips [18,19], dipoles [20,21], and monopoles [22], because the sizes of these devices, as determined by the wavelength, are acceptable and practical for use inside an MRI scanner within this frequency range. Recently, the concept of a loopole coil [23], which is a coil based on creating an unbalanced current distribution through modifications in the capacitor values along the coil, has been proposed. The loopole concept is based on inducing a loop coil to act as a dipole and has been demonstrated to be capable of producing higher field uniformity when used in a volumetric array configuration, and high resistance to loading effects, compared to reference and typical coils [24].

In this study, we present an empirical study by creating a non-symmetric distribution of current on circular coils. In our study, we focus on modifying the position of the capacitor on a circular coil. The circular coil is equipped with a single tuning capacitor, the position of which is changed along the coils in a radial form from its center and is indicated by the angle from the source. The capacitor was rotated from 15° to 345° in steps of 15°, for a total of 23 different configurations. We performed electromagnetic (EM) simulations on coils with capacitor at each position. The simulations were performed on empty space, using a phantom and a realistic human model. We also analyzed the coupling between pairs of coils with non-symmetric capacitor positions by measuring the S_21_ parameters. A total of 24 linear arrays of coils with capacitor position from 15° to 345°, including the reference, were also modelled, with the human spine as the examination target. We also developed a total of 24 pair of loop coils, for which we performed bench tests on the coupling, and obtained images with each coil of a phantom and ex vivo brain with a 7T MRI scanner.

## 2. Materials and Methods

Based on the concept of the loopole, we modified the symmetry of the current distribution using a circular coil with only one tuning capacitor, the position of which was changed along the circumference of the coil. We configured multiple coils and varied the position of the tuning capacitor of the loop coil in intervals of 15°, starting from 15° from the source point, up to 345°. The reference coil [25] was based on a loop coil equipped with tuning capacitors that are equally distributed around the coil, as shown in Figure 1a. The reference coil had three tuning capacitors C_sn_, where *n* = 1, 2, and 3 were located 90°, 180°, and 270°, respectively, from the source. The coil had four segments of conductor line, which can be represented as series resistance R_sn_ and inductance L_sn_. Capacitors and inductors are known for their ability to advance or delay the current in an RLC network. In this study, we want to show that modifications in the current distribution could provide stronger and uniform |B_1_^+^| fields, and that the loop coil could be optimized to have specific field properties depending on the application and region of interest [26,27,28,29,30]. To verify this assumption, we performed numerical simulations [30,31,32,33], a bench test with manufactured coils and MR images with a phantom.

In the simulations and MRI experiments, it was expected that the non-uniform current distribution caused by the non-symmetrical capacitor position in the coil would improve the |B_1_^+^| field, and that this concept can be used to optimize the |B_1_^+^| field for particular applications. Because the amount of data to present was large, we used extracted line profiles, and rather than showing each |B_1_^+^| map, we plotted statistical (mean, standard deviation) values. For simplicity, we labeled the coils based on the angular positions of the capacitors; for example, the coil that had its capacitor at 45° was labeled as C45.

Data analysis was performed using MATLAB (The MathWorks, Inc., Natick, MA, USA), and for data processing, the field maps were resampled to have the same matrix sizes, i.e., to 256 × 256 × 90. We computed the coefficient of variance (CV), which is the standard deviation divided by the mean value, for each case, to determine the uniformity of the field. A lower CV value indicates that the samples are more uniform.

### 2.1. FDTD Simulation

We performed simulations using a commercial finite-difference time-domain (FDTD) software (Sim4life, Zurich, Switzerland). We modeled a circular loop coil that included a source. For the reference coil, we included three tuning capacitors, whereas for each coil configuration that was studied, we included only one tuning capacitor. We performed simulations in empty space and with a loading phantom. For the empty-space simulation, the coils were 100 mm in diameter, whereas for the phantom and human-model analysis, the coils were 80 mm in diameter. The size difference of the coils was set to demonstrate that the concept can be applied to coils of different dimensions. The conductor line had a width of 5 mm, and the material type was set to perfect electric conductor (PEC). The gap between the source and capacitors was set to 3 mm. As mentioned previously, the position of the tuning capacitor along the circumference of the coil was varied from 15° to 345° in intervals of 15°. We used an automatic method to create this model with the aid of a Python script and the Sim4life built-in functions to modify the design parameters. In addition, we constructed a reference coil, which consisted of circular coils with a diameter of 80 mm, wire width of 5 mm, and three evenly spaced tuning capacitors, the positions of which were set to 90°, 180°, and 270°. Figure 1 shows the concept of the coils, a few examples of coils with capacitors at different positions, and the reference coil. Each coil was tuned and matched to 300 MHz and 50 Ω.

#### 2.1.1. FDTD Simulation in Empty Space (Single Coil)

To demonstrate the performance of the concept, we first performed simulations on an empty space (without a phantom). We computed and acquired the current distribution of each coil and compared it with that of the reference and verified the |B_1_^+^| fields. The coils were excited using a Gaussian pulse with a center frequency of 300 MHz and bandwidth (BW) of 600 MHz. The capacitance of the tuning capacitors for the reference coil was 3.1 pF, whereas that for the asymmetric coils was 0.55 pF. It should be noted that, at least for the simulations, the same value of capacitance was used regardless of the position of the capacitor along the coil; however, this behavior changes when there is a loading material. A total of 24 simulations were performed with loop coils placed in an empty space. Each coil was excited to 300 MHz using a Gaussian pulse with a bandwidth of 600 MHz, and all simulations were normalized to an input power of 1 W.

#### 2.1.2. FDTD Simulations Phantom (Single Coil)

We also performed simulations using a phantom. As shown in Figure 2a,b, the phantom had the shape of a cube with 20 cubic subdivisions. This phantom was selected to enable the inclusion of different tissue materials. Furthermore, we set the middle cube to act as cerebral spinal fluid (CSF), shown in blue, and the rest of the cubes as muscle tissue, shown in yellow. The use of the subdivision is also useful for automatic segmentation and region of interest (ROI) assignment. The ROIs were labeled with corresponding numbers, which will be used later for analysis. The separation material between the ROIs was set to be plexiglass. The electrical properties of the CSF (blue box) were set to a conductivity of 2.22 S/m and permittivity of 72.734; the muscle tissue was set to have a conductivity and permittivity of 0.77 S/m and 58.2, respectively [34]. The plexiglass was set to have a conductivity and permittivity of 0.0025 S/m and 2.6, respectively. These values were obtained from a database provided by the software. The phantom had a size of 170 × 170 × 170 mm^3^, whereas each cube had a size of 30 × 40 × 170 mm^3^.

We performed a total of 24 EM simulations with coils with capacitor position from 15° to 345° in the presence of a phantom. The coils had 80 mm in diameter. All simulations were performed using an excitation Gaussian pulse with a center frequency of 300 MHz and a BW of 600 MHz. Each coil was re-tuned and matched to 50 Ω. The distance between the coils and phantom was 10 mm. Individual EM simulations were performed on each coil to acquire the |B_1_^+^| field. Specify absorption rate (SAR) was also computed based on 10 g average, *SAR_10g_*. Normalization of the |B_1_^+^| field was performed via division by the square root of the maximum SAR.
(1)$$Normalized\;\left|B_{1}^{+}\right|=\frac{\left|B_{1}^{+}\right|}{\sqrt{\max\left(SAR_{10g}\right)}}$$

#### 2.1.3. FDTD Simulations Phantom Coil Pairs

With the same phantom, we performed simulations and analyzed the coupling performance of the coil configurations, which were combined in pairs, as shown in Figure 2b. For this analysis, we intentionally positioned two coils such that the distance between the coil centers was 65 mm. The objective was to determine if, at this distance, it is possible to obtain a better decoupling than that of the reference coil. We performed 48 simulations with the following combinations: first, we performed 24 simulations on pairs of coils with the same capacitor positions, and then performed 24 simulations with pairs of coils with capacitors in opposite positions, e.g., 90° and 270°.

#### 2.1.4. FDTD Simulation Human Spine Array Coil

We also performed simulations using a human model, known as the Duke model, provided by the simulation software [35]. This model contains over 100 tissues (including CSF, fat, muscle, bone, white and gray matter, etc.) with electrical properties that correspond to the frequency of the operation of 300 MHz. We used this model to test the coils in the examination of the spinal region by making linear arrays of coils with different capacitor positions. All simulations were performed using a Gaussian pulse with a central frequency of 300 MHz and a BW of 600 MHz, with a normalized input power of 1 W. The simulations involved six pairs of coils: the first pair of coils was located between vertebrae C1 and C3, the second between C4 and C7, the third at T1 and T5, the fourth at T6 and T8, the fifth between T9 and T11, and the sixth at T12, L1, and L2, as shown in Figure 2c,d. Each pair involved all 23 combinations of capacitor positions, for which separate simulations were performed of arrays consisting of coils with capacitor position from 15 to 345°. The diameter of each coil was set to 80 mm. The distance between the pair of coils was selected to be 65 mm from the center of each coil. This distance was selected so that the reference coils would have at least −15 dB of S_21_.

We performed a total of 24 simulations; each simulation was conducted on a coil array consisting of 12 elements, as depicted in Figure 2c,d. Each simulation involved one of the 23 coil configurations with capacitor positions ranging from 15° to 345° in 15° intervals, or the reference coil. For each simulation, the array was composed of coils of the same configuration, e.g., one simulation involved an array consisting only of coils in the C15 configuration, whereas another simulation involved an array consisting only of coils in the C90 configuration. The distance between the centers of the coils in the X direction was 65 mm. Specific absorption rate (SAR) maps were also acquired for each case based on 10g average, SAR_10g_.

### 2.2. Coil Benchwork

The RF coils were developed using a flexible copper material, printed PCB, with special cuts at the capacitor positions. We developed a total of 24 pairs of coils, which are shown in Figure 3a. The diameter of the PCB coil was 80mm and 5mm width, a cut of 4mm was set for the placement of the capacitor. We prepared one pair of coils for each capacitor position from 15° to 345° in 15° intervals, plus a pair of reference coils that had four capacitors evenly distributed. The coils were tuned to 297.3 MHz and matched to 50 Ohms. To simplify tuning and matching, we used variable capacitors. The coils were placed on top of the phantom as the loading condition (Figure 3b) for tuning and matching. We tested the coupling between each pair of coils with the same capacitor positions by measuring the S_21_ parameter [36,37]. We performed the test on three separation distances: 55, 60 and 65 mm between the centers of the coils. Through this test, we intended to demonstrate that the capacitor position in the coil will exhibit better decoupling for different overlapping distances.

### 2.3. MRI Experiment and Phantom Development

We used these coils to acquire MRI images and to study the coupling behavior of nearby coils. The MRI experiments were performed using a 7T MRI scanner (Magnetom, Siemens AG, Berlin, Germany) at 300 MHz. We created a phantom based on a simulation model following the tissue selection of muscle and CSF. We used data from a past study [38] to select the corresponding electrical properties of the phantom. The electrical properties were implemented using a mixture of water, agar, sucrose, and NaCl. Figure 3b shows the phantom and the placement of the coils for analysis.

We tested the performance of the coils by acquiring MR images in the presence of a phantom. The images were obtained using each of the 24 coils equipped with a capacitor at each of the listed angular positions and with the reference coil. The images were acquired using a 7T MRI scanner, with a gradient echo (GRE) pulse sequence, a repetition time of 300 ms, echo time of 4 ms, slice thickness of 3 mm, acquisition matrix of 256 × 256, and flip angle of 30°. We repeated the experiment with each coil using the phantom described in the Method section, for a total of 48 images in transversal and sagittal view.

Flip angle maps, which are a representation of the |B_1_^+^| field, were acquired for the reference coil and the C180, C225 and C245 coils, using the conventional double angle method (DAM) [39,40]. Images were acquired with GRE with FA of 30 and 60°, TR = 1500 ms, TE = 4 ms, matrix of 320 × 320 and slice thickness of 3 mm.

## 3. Results

### 3.1. FDTD Simulation

We performed a study on the effects of the angular position of the capacitor along the circumference of a loop coil. We performed a total of 121 simulations to demonstrate the effects of the capacitor position in a loop coil.

#### 3.1.1. FDTD Simulation in Empty Space (Single Coil)

Figure 4 shows the computed surface current densities (J) of the selected coils. The absolute vector surface current density of the reference coil is shown in Figure 4a. The maximum current density was produced by C15, as shown in Figure 4b; this current density was higher than that of the reference coil. The surface current exhibited two bands of low intensity at 90° and 270°, which were caused by the change in the phase of the x component of the phase of the current density. Figure 4c,d shows the current densities for C90 and C180, respectively. C90 exhibited J with a higher intensity on the opposite side of the capacitor location. Despite the asymmetry of the J distribution, this coil produced the most uniform |B_1_^+^| field in the XY plane. On the other hand, a lower J distribution was computed for C180, which also had the lowest average |B_1_^+^| field. This coil had a J intensity that was lower than that of the reference coil; however, C180 was also a coil with a symmetrical capacitor-wire distribution.

We computed the |B_1_^+^| field in empty space, which depicts slices at the center of the coil in the XY plane (Figure 5a) and in the ZY plane (Figure 5b). To better visualize the performance of each coil, we obtained three-line profiles of the computed |B_1_^+^| field, the legends of the profile lines are included in the top corner of Figure 5a,b. The first profile (white line in Figure 5a, from point L1 to L2) was set 50 mm from the coil, and the results are shown in Figure 5c. The second line profile (Figure 5a, from point L3 to L4) was set at a larger distance, i.e., 200 mm, and the results are shown in Figure 5d. The third line profile is taken from the ZY plane (Figure 5b, from point L5 to L6) 50 mm from the coil, and the results are shown in Figure 5e. These line profiles show that the reference coil (blue solid line) had a less-than-average field strength compared with the other coils. At short and large distances, the coils with capacitors positioned at 15° and 345° exhibited higher field intensities, as depicted by the red solid, and dotted lines. The line profile graphs show an interesting characteristic, where the field intensity varied in accordance with the position of the capacitor: a higher intensity was produced when the position of the capacitor was closer to the source, whereas the lowest value was produced when the capacitor was at 180°. If these values are to be modeled, they will be approximated as a sum of cosine and sine:(2)$$f\left(\theta \right)=a_{0}+a_{1}\cos\left(\theta *w\right)+a_{2}\sin\left(\theta *w\right)$$
where θ is the position of the capacitor given in degrees, and the coefficients $a_{0}$, $a_{1}$, $a_{2}$, and w are dependent on the size of the coil, distance, and orientation of the line profile.

Another important aspect of this graph is how the reference coil compared with other coils; the performance of the reference coil was comparable to those of the coils with the capacitors positioned in the range 105° to 135°. It should be noted that at this point, these inferences have been based only on empty-space simulations; in the presence of a dielectric material (loading case), the field pattern will be affected by the composition of that object. Nevertheless, this empty-space analysis revealed that it is possible to achieve a higher field intensity using coils of the same size and through simple modifications in the position of the capacitor.

In Figure 6, we describe the statistical results for the |B_1_^+^| field in empty space. We summarize the mean |B_1_^+^| field for each line profile. These figures show better correlations between the field intensity and the angular position of the capacitor. At short distances (Figure 6a,c), the reference coil performed better than eight other coil configurations only in terms of field strength. At large distances (Figure 6b), the reference coil performed better than only five of the coil configurations. For visualization purposes, we scaled the CV based on the maximum value for each distance such that the values could be fixed in the same plot as shown in Figure 6d. This analysis is also interesting because it shows that depending on the application or desired field distribution, one can choose a coil configuration that would yield a more uniform field, either at short distances or at long distances. For example, C90 exhibited better field uniformity in the XY plane both at short and long distances from the coil. We also measured the focus of the field produced by each coil, and the difference in distance between the maximum |B_1_^+^| field value and the center of the coil, as shown in Figure 6e. Positive values indicated that the maximum value of the field was oriented to the right, negative values indicated that the field was focused to the left, and a value close to 0 indicated a well-focused field. At short distances in the XY plane, the fields produced by coils C15 and C165 were more focused toward the center. At larger distances, the field produced by the coil with its capacitor at 90° was more focused toward the center compared to the fields produced by the other coils.

#### 3.1.2. FDTD Simulations Phantom (Single Coil)

Figure 7 shows the |B_1_^+^| field for the selected coil configurations. We compared the |B_1_^+^| field in the XY plane at the center of the coil between the reference coil (Figure 7a left) and C315 (Figure 7a right). We used the marked green line (points L1 to L2) to create a field line profile, such that it would be easier to compare the coils with each other, as shown in Figure 7c. The reference coil is indicated by the blue line. This graph shows an interesting pattern: the coils with their capacitors positioned between 15° to 150° exhibited higher field intensities focused on the left part of the field (negative x), with the highest intensity achieved by the coil with its capacitor at 75°. The coils with capacitors at 15° and 150° exhibited field intensities in this region similar to that of the reference coil. The coils that performed better in the right part of the field (positive x) were those with capacitors between 210° and 345°, with the highest field intensity in this region produced by the coil with its capacitor at 285°. The coil configuration that produced the best mean value for this line profile was C315. This indicated that it is possible to control the field focus through simple changes in the capacitor position. The |B_1_^+^| fields in the ZY plane for the reference coil (left) and C315 (right) are shown in Figure 7b, respectively. The line profile (points L3 to L4) in Figure 7d shows that C285 and C270 exhibited the highest field intensities, whereas the reference coil exhibited a field intensity that was slightly better than the average among all the coils.

A statistical analysis of the simulations involving the phantom is shown in Figure 8. The mean and CV computed for the whole phantom (volume) for each coil configuration are shown in Figure 8a,b, respectively. This analysis revealed that coil C315 exhibited the highest field intensity, and that a total of 10 coil configurations resulted in better field intensities than that of the reference, with the lowest value achieved using C150. On the other hand, coil C240 resulted in the lowest standard deviation; moreover, eight coils resulted in better standard deviations than that of the reference coil. We segmented and created ROIs at each rectangle of the phantom to visualize the field patterns for each coil configuration. Figure 8d shows a representation of the mean value at each ROI for each coil configuration. The ROI number is shown on the *y*-axis, whereas the coil type is shown on the *x*-axis. We normalized each ROI (row) based on the maximum mean value among the coils to make it easy to visualize which coil had the best performance at each ROI. We used a heat-color map to visualize these values, with a value of 100 representing the maximum field intensity on that row. This type of analysis could be used for designing loop coils with specific target zones or patterns, and easy to spot the higher field intensities per ROI. For example, the coil configurations from C15 to C150 exhibited higher field intensities toward the left, whereas the fields of C210 to C330 were more focused toward the right; this result also demonstrated that higher field penetration can be achieved using coil configurations C270 to C315. Of particular interest is ROI number 8, which was assigned electrical properties simulating those of CSF; the mean values at this ROI for all tested coil configurations are shown in Figure 8c. In this ROI, the coil with its capacitor positioned at 285° exhibited the highest mean value, which was 17% higher than that of the reference coil. This coil configuration was also demonstrated to be capable of producing the highest field intensity over a long distance from the coil, with most of the maximum values in the ROIs from 15 to 20, corresponding to the last row of the phantom.

When a transmission coil is analyzed, it is often desirable to show the SAR, and the performance of the magnetic field in relation to the SAR. In Figure 9a, we show the 10 g-averaged SARs computed for the reference coil (left) and coil C180 (right), respectively.

Figure 9b shows the normalized |B_1_^+^| fields for the reference and coil C345, respectively. Figure 9c shows the maximum SAR for each of the coil configurations. The lowest SAR was exhibited by the C180, whereas the highest was exhibited by the C60 coil. The mean value of the normalized |B_1_^+^| field is shown in Figure 9d. The results show that coil C345 exhibited the best performance, especially when compared with that of the reference coil.

#### 3.1.3. FDTD Simulations Phantom Coil Pairs

We also performed EM simulations on combinations of two coils (as depicted in Figure 2b) to analyze the performance in terms of coupling, as shown in Figure 9e,f. Figure 9e shows the S_21_ parameters between coils with the same capacitor positions. As shown in the figure, the coils with capacitors positioned between 135° and 225° exhibited better decoupling than that of the reference coil. Specifically, coil C180 exhibited an S_21_ of −22.26 dB, whereas the reference coil exhibited an S_21_ of −17 dB. On the other hand, for the coils with capacitors in opposite positions, including the pairings with coils C90 and C180, the S_21_ parameters are shown in Figure 9f. For this type of configuration, coils C195 and C165 resulted in better decoupling than those of the other combinations; however, the values were still lower than that of the coil pair with C180.

#### 3.1.4. FDTD Simulation Human Spine Array Coil

EM simulations with a male human model were performed with the coil array consisting of 12 elements, as depicted in Figure 2c,d. Each simulation involved one of the 23 coil configurations with capacitor positions ranging from 15° to 345° in 15° intervals, or the reference coil. Figure 10a–d show the |B_1_^+^| field normalized by the square root of the maximum SAR. For the |B_1_^+^| field in the XY plane, that of the reference coil array (Figure 10a) is shown in the top row, whereas that of the C75 coil array (Figure 10b) is shown at the bottom row; Figure 10a,b show the XY planes at the centers of each pair of coils, as illustrated by the dotted lines in the ZY plane of |B_1_^+^| field in Figure 10c,d. Coil C75 exhibited a higher field intensity than that of the reference coil array, both in the XY and ZY planes.

In Figure 10e,f, we summarize the statistical analysis for the entire body volume inside the simulation and resize the matrix sizes of the simulations such that they are depicted with the same dimensions. Figure 10e shows the |B_1_^+^| field mean value for each of the coil arrays. The C75 coil array exhibited a higher field intensity than those of the other coil arrays, whereas the reference coil array exhibited the lowest field intensity. Furthermore, C180 exhibited the lowest mean value among the coils equipped with a single capacitor; incidentally, C180 was also a symmetrical type of coil. With regard to the uniformity, we computed the CV for each coil array and plotted the corresponding values in Figure 10f, which shows that C285 was the coil array that produced the most uniform field, which was approximately 60% better than that produced by the reference coil array. We also computed the maximum SAR_10g_ for the whole body, as shown in Figure 10g, which shows that C60 and C300 had the highest SAR10g values, and that C180 had the lowest maximum SAR_10g_. The results show that, in terms of the SAR, at least six of the coil arrays performed better than the reference coil array.

As an example of how this study could be applied to coil array design, we present a simple coil selection with the objective of improving the |B_1_^+^| field distribution on the spinal cord. The goal of this selection was to obtain a similar field intensity along the entirety of the spinal cord. The reason for this objective is that when a planar linear coil array is used, the field intensity is higher at the regions of the body that are closer to the coil, usually at approximately T6 and T11 of the thoracic area of the spinal cord, whereas the cervical area C1 to C4 would be subjected to lower field intensities because of the larger distance to the coils. The |B_1_^+^| field along the spinal cord could be optimized to be of a similar amplitude through selection of coil configurations that would yield a similar field intensity at every region of the spinal cord. To demonstrate this example, we performed a statistical analysis for each XY plane marked. We divided each region with ROI S_i_ and computed the mean |B_1_^+^| field for each of the coil arrays, the results of which are shown in Figure 11a. In this figure, the *y*-axis identifies the six regions corresponding to the position of the pair of coils, whereas the *x*-axis identifies the type of coil. For region S_1_, which corresponds to cervical segments C1 to C4 of the spinal cord, the coil that exhibited the highest mean field value was C75, with an average of 0.033 μT. If we use this value as a reference, we can determine other coils that would have similar mean field values in the other regions. For regions S_2_ to S_6_, we can select the pairs of coils with C60, C240, C150, the reference coil, and C75, respectively. Each of these coils would have an approximate field value of 0.033 μT. We performed EM simulations using this selection of coil pairs to create an array. The resulting |B_1_^+^| field is shown in Figure 11c, including a comparison with the reference coil array (Figure 11b). In this field map, to show the field improvement from the coil array we highlighted the value (green contour) of the field equal to 0.02 uT, it can be seen that this value was more consistent along the spinal cord for the case of the optimized array of selected coils in comparison with the reference coil array. The field standard deviations computed for only the spinal cord for the reference and optimized coil arrays were 0.024 and 0.010 μT, respectively. This simple example demonstrated that it was possible to enhance the field pattern through the selection of coils with different capacitor positions.

### 3.2. Coil Benchwork

The S_21_ parameters are shown in Figure 12a–c. We performed the test on three separation distances: 55, 60 and 65 mm between the centers of the coils, as shown in the top, middle, and bottom panels, respectively. The column of Figure 12a shows the S_21_ parameters for the reference coil, whereas the column of Figure 12b shows those for C45. Figure 12c shows a summary of the S_21_ parameters for all the capacitor position in the coil. According to the figure, for a distance of 55 mm, which corresponded to a high overlapping distance, the coils with the capacitor at 315° exhibited a low coupling of approximately −30 dB, in comparison to the reference coil, which had a coupling of −7 dB coupling. Most of the coils exhibited a lower coupling than that of the reference coils. When the distance between the coils was 60 mm (middle row in the figure), coils C45 and C255 exhibited a coupling of −20 dB, whereas when the distance was 65 mm (lower row), coil C30 had a coupling of −17 dB. In addition, we observed a number of special cases, where coils C105 and C285 exhibited couplings of −21 and −27 dB, respectively, when the distance between the coupled coils was 50 mm. Meanwhile, C300 had a coupling of −24.9 dB when the coil separation distance was 45 mm, whereas C90 exhibited a coupling of −20.6 dB when the coil separation distance was 40 mm.

### 3.3. MRI Experiment

The images acquired using the selected coils are shown in Figure 12d–g, showing the axial and sagittal views, respectively. We performed a comparison between the reference coil (Figure 12e), C45 (Figure 12f), and C240° (Figure 12g). The images are visualized with the same intensity window. For the reference coil, the images for ROIs 2 and 4 have lower intensities than those for C45. Meanwhile, the bottom row of the image, depicting the observations for C240, shows a higher intensity compared with those of the other two coils. We examined the image SNR in each ROI of the phantom and visualized the summarized values in Figure 13, where the *x*-axis denotes the coil type, starting with the reference, and the *y*-axis denotes the ROI region. We used a heatmap representing the SNR values; for easy visualization, we normalized the values to the maximum for each ROI, such that a row or ROI will show which coil had the best performance. According to the figure, the coil with its capacitor at 45° exhibited higher SNR values than those of the other coils, whereas coil C240 also exhibited high intensity in its ROIs, according to the lower row of the figure. The statistical analysis for the sagittal view is shown in Figure 13b, which visualizes four regions and shows that C240 exhibited the highest SNR value.

The flip angle maps are illustrated in Figure 14 for the selected coils. Two lines were selected for analysis: the first line, labeled L1, is located in the first row of the phantom, and the profile plot is shown in Figure 14e. The second line, L2, is located at the ROI 8, and the profile plot is shown in Figure 14f. The mean and standard deviation for the line profiles, L1 and L2 are displayed in Figure 14g,h, respectively. These results indicate that the C225 produces a higher field intensity than the reference coil. The same coil also showed better uniformity with lower standard deviation in the selected L1 and L2 locations. The standard deviation of C225 in L2 showed 4.6 times better uniformity than the reference coil. In terms of peak intensity, the C180 showed the maximum intensity.

A summary for the best configuration and comparison with the reference coil is present in Table 1, showing the coils with best performance for each of the tested parameters.

## 4. Discussion and Conclusions

We presented a study on the effects of changing the capacitor position in a loop coil, such that the findings can be applied to high-frequency applications in a 7T MRI scanner. We performed EM simulations and verified the simulation results using MR images of a phantom. We also fabricated the coils and bench-tested the couplings of different pairings of coils. This study can be beneficial for applications where the |B_1_^+^| field should be optimized, or the receiving coil arrays should be characterized by different field patterns or focus the |B_1_^+^| field into a specific imaging region, or the SAR input into the patient is supposed to be reduced. This research can also be extended to higher frequencies and to the development of volumetric coil arrays. The concept of changing the capacitor position to alter the current distribution, as demonstrated in this study, also implies that breaking the symmetry of the resonators could enable the realization of target field patterns and be used as a control mechanism for optimizing B1 field intensity, uniformity, reducing SAR or improving coupling between coils.

In this study, we have presented that by changing the capacitor position along the coil, it could provide better performance than the reference loop coil. For the study cases presented in this work (coil size, frequency, phantom models, etc.), the simulations indicate that a coil with a capacitor at 15 or 345°, would provide a 35% higher field intensity than the reference coil for the case in empty space. The EM simulations with a phantom showed that the coil with capacitor at 285° had 17% higher field intensity than the reference coil. The SAR simulations also showed that the coil with capacitor at 180° had 10% less SAR than the reference coil. In terms of best transmission efficiency, normalizing the |B_1_^+^| field with the SAR, the coil with capacitor at 345°, had a 4% improvement over the reference coil. In terms of coupling between the adjacent coils, the simulations showed that coils with capacitor positions from 150 to 215° had S_21_ of at least −22 dB compared to the −17 dB of the reference coil. These results are in agreement with the concept of self-decoupling coil [26], in which a higher impedance is located opposite to the source in a loop coil. EM simulations with a spine array with coils of capacitor position of 75° had a field intensity 60% higher than an array of the reference coils.

When performed benchtop measurements, the coupling between two adjacent coils with capacitor position at 45° and separated by a distance of 60mm had a S_21_ of −20 dB, while a pair of reference loop coils had a S_21_ of −11 dB, at the same separation distance. Flip angle maps show that the coil with capacitor position at 225° has a higher field intensity, and field uniformity was 4.6 times better compared to the reference coil.

This study demonstrated that the effect of the capacitor position along a loop coil can be considered for optimizing the designs of either transmission or reception coils for ultra-high frequencies. In this study, we performed simulations and experiments only for the operation frequency of 300 MHz; however, a similar study could be applied for higher frequencies that are used in MRI systems with a stronger main magnetic field (B_0_). Similarities between the mean values were observed in the simulations involving empty space, phantom, and spine array. The results in the presented plots show that, in the case of coil C180, the field intensity is minimum when the capacitor is near symmetry, whereas the field intensity was higher when the capacitor was closer to the source. Although this pattern changes depending on the loading condition, the graphs show a general concept of what can be expected. Similarly, the SAR values indicate that C180 can produce a lower SAR. There is also a relationship between the observations obtained from the simulation of couplings between two coils and the bench test on the developed coils, which reveal two local minima and one global minimum. Furthermore, the coils with the capacitors closer to the source demonstrated half of the dBs of the coils around C180, with the coils around C90 and C270 exhibiting the lowest dBs. This pattern also depends upon the fabrication of the coils, such as the vertical distance between the coils and matching board circuits. Nevertheless, this study suggests that higher overlapping distances can be achieved through simple changes in the capacitor position in the loop coil. This concept can also be applied to rectangular coils; however, in that case, the loopole concept may be more practical. The present work demonstrates a simple method to achieve similar results as the loopole [23] and the self-decoupling [27] coil but with a single capacitor. The present work and the previous works have in common that the current distribution is controlled in a non-symmetrical manner, therefore our results are in alignment with previous work and theory. However, in [27], the work was focused to address the decoupling between coils, and in [23], the objective was to find uniform |B_1_^+^| field, whereas in our work, we demonstrate that coupling between coils, |B_1_^+^| field and SAR, can be improved by simply changing the capacitor position.

However, while the position of a single capacitor can enhance the performance of a loop coil, it also has some limitations; for example, to find the optimal position requires an update of the model of the coil for each simulation. The coil model needs to change the capacitor gap and the capacitor itself. However, the modeling process could be automated by using scripts to create the model. Another limitation is to develop the coil, such that the geometry and the capacitor gap should be correctly designed. In the present work, we used printed PCB with the specific geometry and capacitor gap, for better accuracy of the capacitor gap position.

We hope that the current study can demonstrate the application and concept that changing the capacitor position can improve either, |B_1_^+^| field, SNR, SAR, or coupling. The position of the capacitor will ultimately be set based on the imaging target, arrangement and application of the designer.

## Figures and Tables

**Figure 1 biosensors-12-00867-f001:**
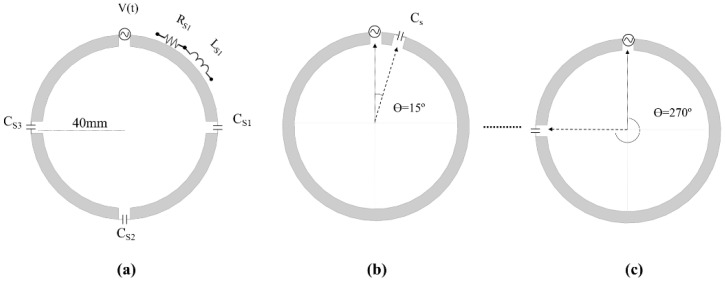
Visualization of concept of non-symmetric distributed RF coils, with (**a**) reference, and coils with capacitor positions at (**b**) 15° and (**c**) 270°. Models for EM simulations involving.

**Figure 2 biosensors-12-00867-f002:**
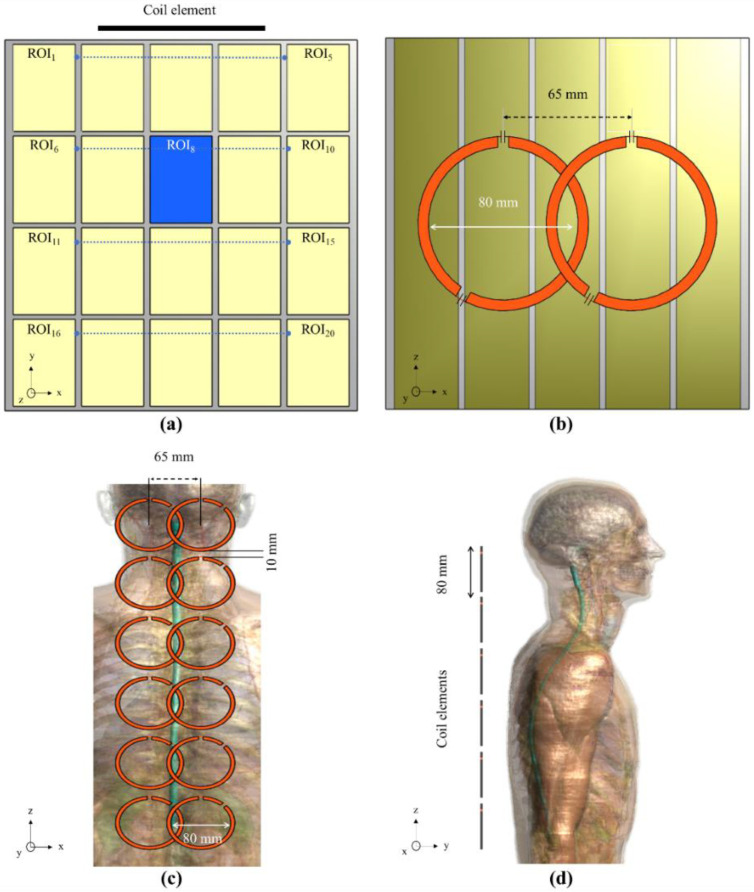
(**a**) Phantom, indicating position of each ROI, (**b**) setup used for coupling analysis between two coils of the same type. Array consisting of 6 pairs of coils for human model, with spinal cord as the target, in (**c**) XZ plane and (**d**) YZ plane.

**Figure 3 biosensors-12-00867-f003:**
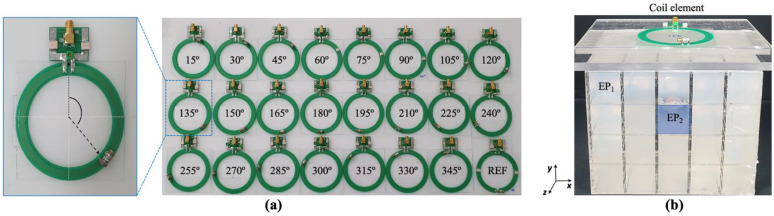
RF coils created for this study, showing (**a**) single coil with capacitor placed at 210° with corresponding matching circuit, and 24 coils, including reference coil, for coil combinations with capacitor positioned between 15° and 345°, in 15° intervals. (**b**) ROI phantom with 20 ROIs, and location of coil on top for MRI imaging.

**Figure 4 biosensors-12-00867-f004:**
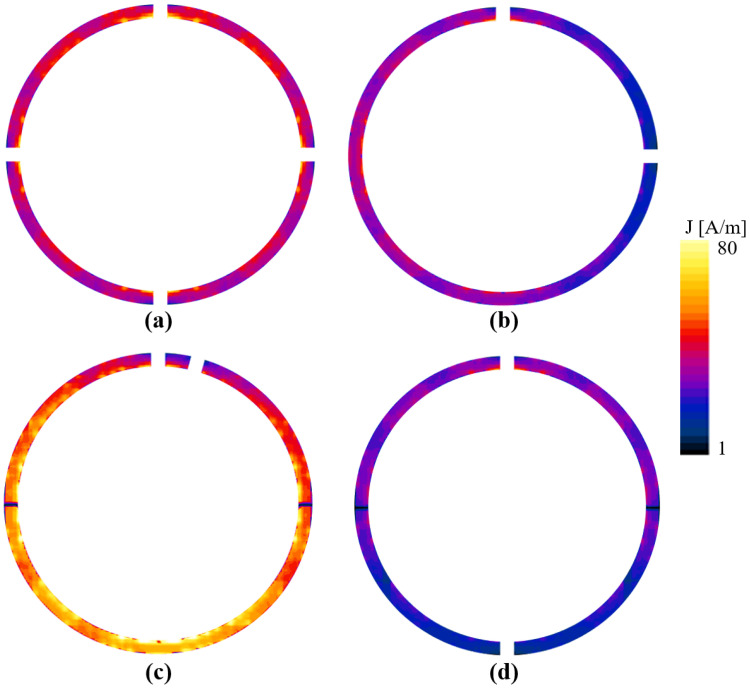
Current densities of (**a**) reference coil, and of coils with capacitor positioned at (**b**) 15°, (**c**) 90°, and (**d**) 180°.

**Figure 5 biosensors-12-00867-f005:**
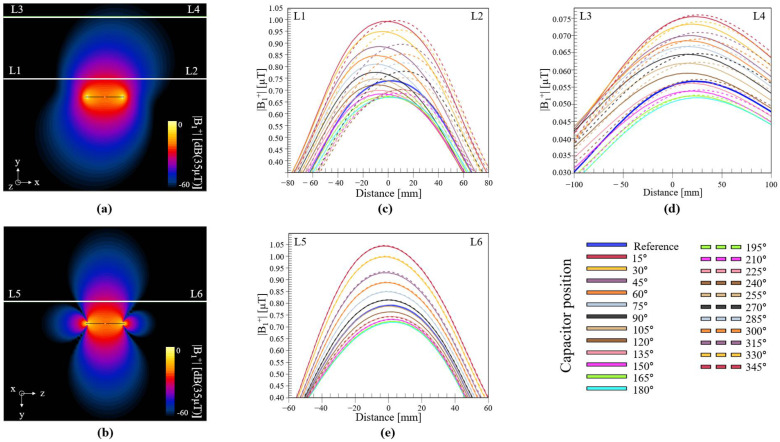
Computed |B_1_^+^| field in empty space in (**a**) XY plane and (**b**) YZ plane. (**c**) Plots of L1–L2 profile lines, (**d**) L3–L4 profile lines, and (**e**) L5–L6 profile lines, for all coil configurations.

**Figure 6 biosensors-12-00867-f006:**
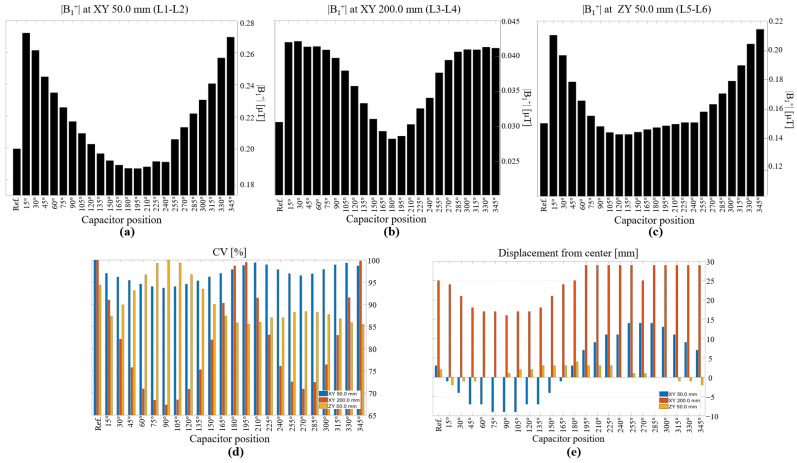
Statistical analysis of |B_1_^+^| field in empty space, showing mean values for line profiles (**a**) in XY plane, at 50 mm, (**b**) in XY plane, at 200 mm, and (**c**) in ZY plane, at 50 mm. (**d**) CVs for all tested coil configurations and line profiles; values have been scaled based on the maximum for easier comparison. (**e**) Focus of the field, computed based on displacement of maximum value from the center, for all line profiles.

**Figure 7 biosensors-12-00867-f007:**
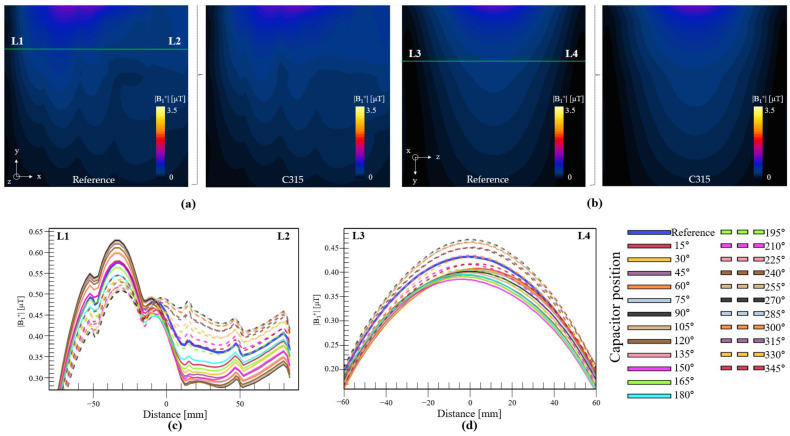
Computed |B_1_^+^| field inside phantom in XY plane for (**a**) the reference coil and the coil with capacitor at 315°. The field in view in the YZ plane (**b**) for the reference coil, and coil with capacitor at 315. (**c**) L1–L2 line profile and (**d**) L3–L4 line profile.

**Figure 8 biosensors-12-00867-f008:**
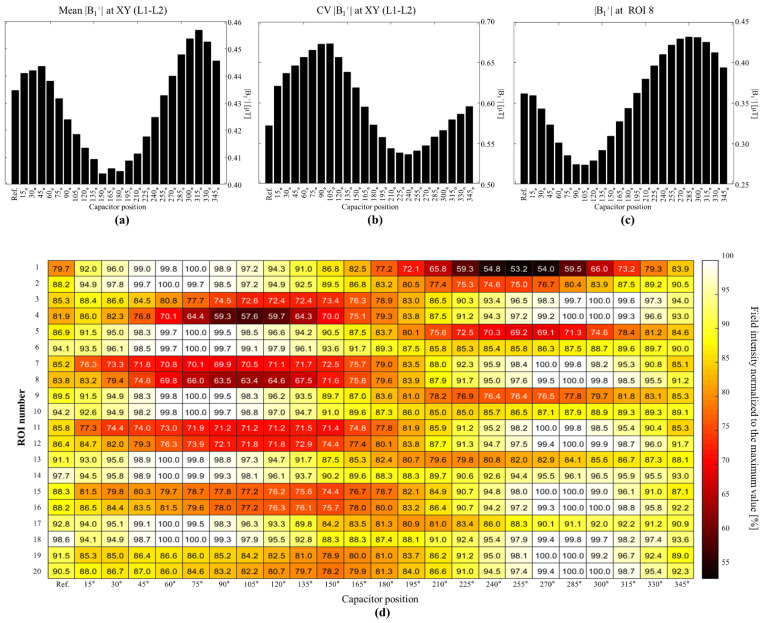
(**a**) Mean values of |B_1_^+^| field in the XY plane, (**b**) CVs in the XY plane, (**c**) mean values of |B_1_^+^| field in ROI 8 for all tested coil configurations. (**d**) Mean values in all ROIs of phantom for all tested coil configurations.

**Figure 9 biosensors-12-00867-f009:**
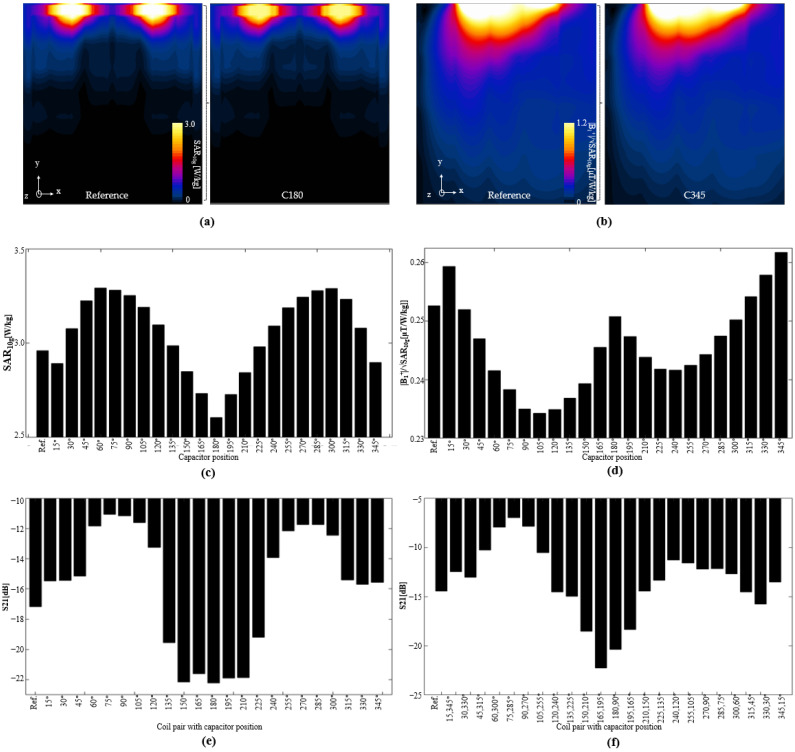
Computed SARs in XY plane for (**a**) reference coil, and coil with capacitor at position 180°. Normalized |B_1_^+^| field based on maximum SAR (**b**) for the reference coil and coil with capacitor at position 345°. (**c**) Maximum SARs in phantom for all tested coil configurations, and (**d**) mean values of normalized |B_1_^+^|/SAR_max_ field for all tested coil configurations. Coupling analysis of S_21_ parameters for (**e**) coils with identical configurations, and (**f**) coils with opposite capacitor configurations.

**Figure 10 biosensors-12-00867-f010:**
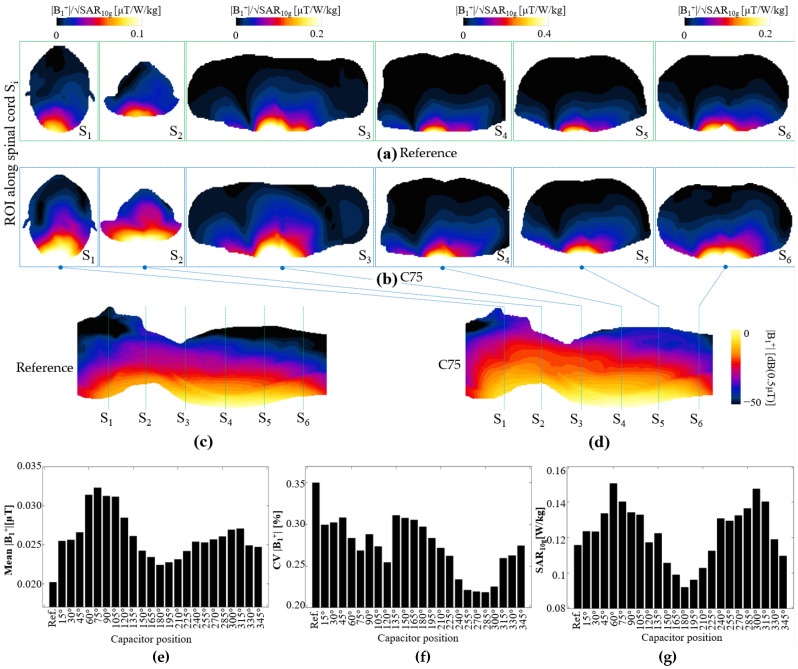
Computed |B_1_^+^| field in XY plane for arrays of (**a**) reference coils, with details shown in top row, and (**b**) coil with capacitor at position 75°, with details shown in bottom row. Computed |B_1_^+^| field in the ZY plane for the (**c**) reference array, and (**d**) the array with capacitor at position 75°, the selected ROIs are labeled as S_i_. Statistics of computed |B_1_^+^| field: (**e**) mean, (**f**) CV, and (**g**) maximum SAR_10g_ values for all tested coil configurations.

**Figure 11 biosensors-12-00867-f011:**
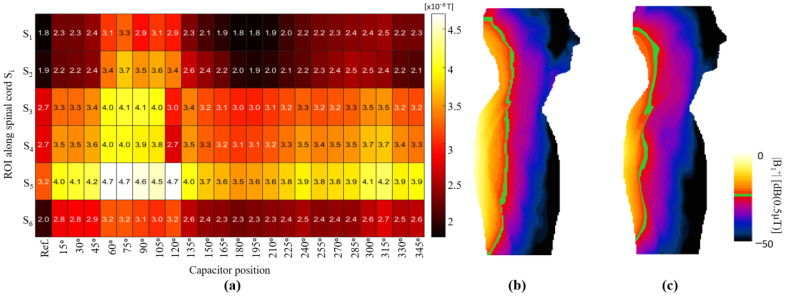
(**a**) Mean values of |B_1_^+^| field for all slices at coil center for all tested coil configurations. |B_1_^+^| field in ZY plane for (**b**) reference coil array, and (**c**) optimized coil array, showing highlighted magnetic field.

**Figure 12 biosensors-12-00867-f012:**
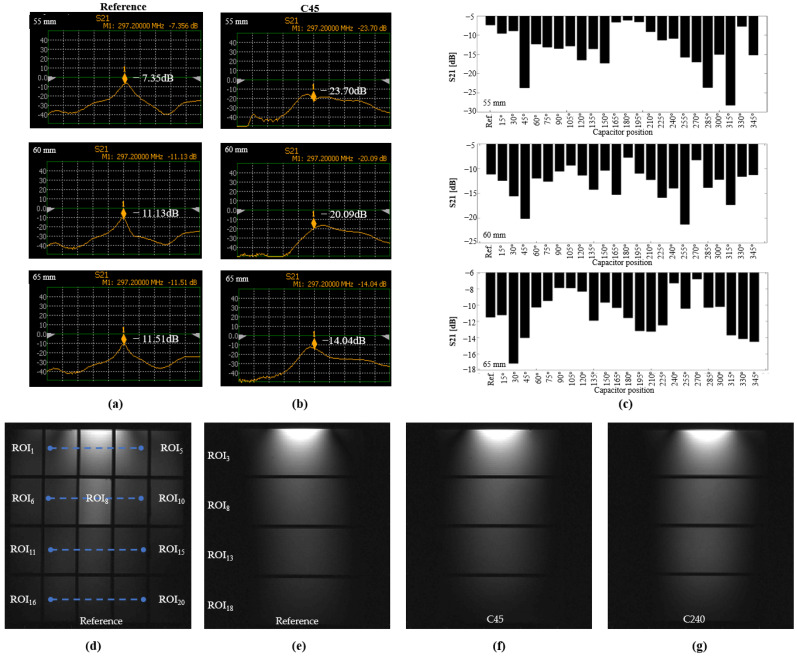
(**a**) S_21_ parameters from network analyzer for reference coil, (**b**) C45, and (**c**) S_21_ parameters for all tested coil configurations at 297.3 MHz, for coil separation distances of 55, 60, and 65 mm, shown in top, middle, and bottom panels, respectively. Acquired GRE MRI images in axial (**d**) and sagittal views, for (**e**) reference coil, and coils with capacitors at (**f**) 45° and (**g**) 240°.

**Figure 13 biosensors-12-00867-f013:**
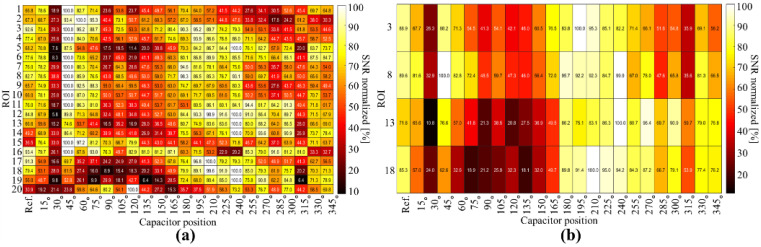
Mean normalized SNR values for each ROIs and for all tested coil configurations in (**a**) axial and (**b**) sagittal views.

**Figure 14 biosensors-12-00867-f014:**
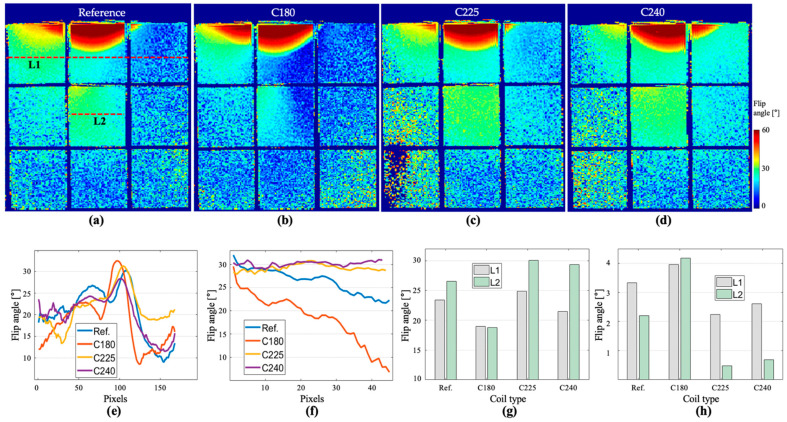
The computed flip angle maps for the (**a**) reference coil, (**b**) C180, (**c**) C225 and (**d**) C240 coils. The line profile for (**e**) L1 and (**f**) L2. The (**g**) mean value of the line profiles, and the (**h**) standard deviation for each line profile.

**Table 1 biosensors-12-00867-t001:** Comparison of the reference coil with the best capacitor position for each of the test conditions.

Test Scenario	Reference	Coil Type	Coil Type Value	Difference
Empty space |B_1_^+^| field single coil	0.19 μT	C15	0.27 μT	42%
Phantom |B_1_^+^| field single coil	0.36 μT	C285	0.43 μT	19.4%
Phantom SAR_10g_ single coil	2.9 W/kg	C180	2.6 W/Kg	−10.3%
Phantom transmit efficiency B1/SAR single coil	0.251 μT/W/kg	C345	0.263 μT/W/kg	4.8%
Phantom coupling S_21_ pair coils	−17 dB	C180	−22.2 dB	30.5%
Human model |B_1_^+^| field array coil	0.020 μT	C75	0.032 μT	60%
Human model SAR_10g_ array coil	0.118 W/kg	C180	0.09 W/kg	−23.7%
Benchwork measured coupling S_21_ pair coil	−11.8dB @65mm	C315	−27dB @ 55mm	128.8%
MR image SNR	82.7%	C45	100%	20.91%
MR image flip angle	26°	C225	30°	15.4%

## Data Availability

Not applicable.

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
