# Peer review of "Study on the Effect of Non-Symmetrical Current Distribution Controlled by Capacitor Placement in Radio-Frequency Coils for 7T MRI"

_biosensors, 2022, doi:10.3390/bios12100867_

Round 1
Reviewer 1 Report
Dear Author,
In your paper, you present a study on the effects of varying the position of a single tuning capacitor in a circular loop coil as mechanism to improve the coupling between adjacent coils, optimize transmission field uniformity or intensity, improve signal-to-noise ratio (SNR) or specific absorption rate (SAR). The treated topic is very interested and I realize that you have done a hard but complete work. The methodology is well described as well as the results; I have just some suggestions to improve the structure of the manuscript, for the benefit of its readability.
General comments:
1) Several parts in the results should be moved to the materials and methods section. Moreover, some parts in the results should be moved to discussion section. I indicate some of these parts in Specific comments but, please, check carefully the entire Results section
2) I think that the results about the simulation at 100Mhz are not important for the manuscript purpose and does not add any useful information so I suggest to remove or, whenever you want report them, please improve the discussion about this point.
3) I suggest to modify the names of subtitle in Materials and Methods section to better separate FDTD simulations from MRI experiments. For example (but this is only a suggestion), you can use: 2.1 FDTD Simulations, 2.1.1 Empty space (unloaded coil) 2.1.2 Phantom and Human Spine; 2.2 MRI Experiments, 2.2.1 Phantom, 2.2.2 Ex-vivo Brain. Then, in the Results section, you should follow the same order to present the results. Finally, you should better highlight when the single loop coil or array were used.
4) I have a doubt about the importance of MRI Experiment on ex-vivo brain: I think that this experiment is not very suitable since your simulations involve a phantom, and a human spine model…
Specific comments:
1) In the Materials and Methods section (FDTD simulations) please, add the more details about the simulations i.e. specific characteristics of PC used, and the mean length of them,… (I am very curios about the total number of simulations that you performed). Please, add some details about the estimation of SAR (i.e. is a SAR1g or SAR10g? )
2) Several parts in the results should be moved to the materials and methods section, for example:
Lines 177-186 and 188-189
Lines 274-280
Lines 337-340 and Equation 2
Lines 354-360 and 377-386
Lines 442-449 and 450-453
Lines 474-480
Lines 504-506 and 517-519
Some parts in the results should be moved to discussion section, for example:
Lines 239-244 and 286-294
Lines 317-328
3) Please, add in the Results a summary (maybe a table) with the indication of the best configuration of coil (capacitor position for single loop and combination of coils for array) vs the mentioned parameters (B1, SAR, S21, uniformity….). This make the manuscript complete and very useful.
4) Figures:
Fig. 3: I suggest to add the axes to better understand the orientation of the phantom and the coil (also respect to the MRI scanner, i.e. B0 direction)
Figure 7: I suggest to add the indication of the coil type (ref or C315) just under the field maps, for easy and quick understanding of the figure (also for Figure 9, 12, 14…)
Figure 11: please, clearly explain what the green profile represents
Figure 13: please, check the caption and add what you mean with Mean values…(i.e. Mean values of Normalized SNR)
5) References:
I suggest to add a citation for the reference coil (line 76), for example a generic review such as Giovannetti et al Critical Reviews™ in Biomedical Engineering, 42(2):109–135 (2014)
I suggest to add a citation for the flip angle used as representation of the |B1 +|-field (line 504) for example: Hartwig et al. Magnetic Resonance Imaging 29 (2011) 717–722
Reviewer 2 Report
In this paper, the authors investigated the effects of altering a surface current distribution on circular coil designed for operation at 7T MRI. The position of single tuning capacitor was varied from 15 to 345 degrees in steps of 15 degrees. The motivation is to improve coupling, transmission field density and uniformity, to improve SAR and SNR.
1. In MRI settings, especially at the UHF MRI, the RF cols are placed as close as possible to the region of interest in order to maximize transmit efficiency and receive sensitivity. For that reason, free space case is irrelevant for MRI conditions.
2. In this paper, the couple of coils with the same position of capacitor were investigated and additionally the coils were overlapped partially. By partial overlap of the coils, it is possible to reduce coupling between them, so we don’t know exact influence of the asymmetrically positioned capacitor (for different position of capacitor, different amount of overlap can be needed). Also, the case of the pairing the coils with different position of capacitor have not been investigated (for example coil with C15 and C195).
3. It is shown that different positions of the capacitor produce different current distributions on a coil, what can be concluded from that knowledge?
4. Also, in this paper coupling between two coils have been investigated but not if the coils are placed in dense array configuration, which is common coil configuration at the UHF. In that case, the coil coupling between the neighboring as well as non-adjacent coils should be investigated. It is very important, for example, in a case of construction of array for brain imaging.
5. If the coil array is proposed for spine imaging, then phantom simulation, phantom measurements, voxel model (Duke) simulations and eventually in-vivo measurements should be performed. The parameters to be observed are B1+, SAR and B1+/sqrt(SAR).
6. The post-mortem head have been measured while being in saline solution. That measurement is not good since the saline solution is giving a very strong signal and the brain imaging is corrupted. The proper measurement can be done if only brain is fixed in formalin and as such placed in a scanner and scanned. After being scanned with different coils, the SNR and B1+ maps could be evaluated and compared.
Round 2
Reviewer 2 Report
no further comments